# Polyvinylidene Fluoride-Graphene Oxide Membranes for Dye Removal under Visible Light Irradiation

**DOI:** 10.3390/polym12071509

**Published:** 2020-07-07

**Authors:** Sabri Alyarnezhad, Tiziana Marino, Jalal Basiri Parsa, Francesco Galiano, Claudia Ursino, Hermenegildo Garcìa, Marta Puche, Alberto Figoli

**Affiliations:** 1Department of Applied Chemistry, Faculty of Chemistry, Bu-Ali Sina University, Hamedan 65174, Iran; sabri_alyarnezhad@yahoo.com (S.A.); parssa@basu.ac.ir (J.B.P.); 2Institute on Membrane Technology, CNR-ITM, Via P. Bucci 17c, 87030 Rende (CS), Italy; f.galiano@itm.cnr.it (F.G.); c.ursino@itm.cnr.it (C.U.); 3Instituto Universitario de Tecnología Química (CSIC-UPV), Universidad Politecnica de Valencia, AV. de los Naranjos s/n, 46022 Valencia, Spain; hgarcia@qim.upv.es (H.G.); mpuche@itq.upv.es (M.P.)

**Keywords:** PVDF-GO membranes, photocatalytic membranes, triethyl phosphate, GO nanosheets, dye removal, photocatalysis

## Abstract

In this study, polyvinylidene fluoride (PVDF)-graphene oxide (GO) membranes were obtained by employing triethyl phosphate (TEP) as a solvent. GO nanosheets were prepared and characterized in terms of scanning and transmission electron microscopy (SEM and TEM, respectively), atomic force microscopy (AFM), X-ray photoelectron spectroscopy (XPS), chemical analysis and inductively coupled plasma mass spectroscopy (ICP). Two different phase inversion techniques, Non-Solvent Induced Phase Separation (NIPS) and Vapour-Induced Phase Separation (VIPS)/NIPS, were applied to study the effect of fabrication procedure on the membrane structure and properties. Membranes were characterized by SEM, AFM, pore size, porosity, contact angle and mechanical tests, and finally tested for photocatalytic methylene blue (MB^+^) degradation under visible light irradiation. The effect of different pH values of dye aqueous solutions on the photocatalytic efficiency was investigated. Finally, the influence of NaCl salt on the MB+ photodegradation process was also evaluated.

## 1. Introduction

The shortage of fresh water and increasing industrial discharge from human activities are the most critical concerns in industrialized cities. In this framework, several processes oriented to the treatment and reuse of wastewaters and seawater such as membrane bioreactors (MBRs) [1], membrane distillation [2], solar to steam generation systems [3,4,5] are undergoing rapid expansion.

One of the most important sources of water contamination is the textile wastewater that includes a variety of organic dyes, which should not be underestimated because of their poisonousness, carcinogenic potential and non-biodegradability. From this perspective, effective wastewater treatment technologies with minimum consumption of energy and reagents for chemical conversion of organic dyes into safe (non-toxic) or harmless compounds are required [6]. Photocatalytic membranes, including inorganic [7,8,9,10,11] and polymeric membranes [12,13] under UV [9,14,15,16,17,18], visible [19] and UV-visible [8,20,21] irradiations have been studied. Because of its harmfulness and high-energy consumption, UV irradiation is undesirable. Being ecofriendly and natural, visible light has shown great potential for energy-efficient water purification. In fact, only 4–7% of the solar spectrum is UV and 47% of that is visible light irradiation [8]. Moreover, the majority of polymeric membranes cannot tolerate UV light for long irradiation times, hence expensive ceramic membranes should be preferred, with an increase in capital costs [22]. Coupling visible-light photocatalysis and membrane separation technology in a single stage could simplify the water treatment process in accordance with the strategy of Process Intensification [23]. Membrane technology has been used due to its high selectivity, easy control, operation and scale-up, low environmental impact and low energy consumption. However, the use of volatile and toxic organic solvents, such as N-methyl-2-pyrrolidone (NMP), N,N-dimethylformamide (DMF) and N,N-dimethylacetamide (DMA) which represent the major components during the phase inversion procedure, does not allow most of the current membrane production processes to be defined as “green”. On the contrary, triethyl phosphate (TEP), being completely miscible with water and possessing a high boiling point (215 °C), represents a non-toxic solvent, already studied for polyvinylidene fluoride (PVDF) membrane fabrication in safer operational conditions [21,24,25]. The proper TEP solvency power towards PVDF is also confirmed by thermodynamic factors, that is, Hansen solubility parameters, which clearly evidence that homogeneous PVDF-TEP solutions can be easily obtained [26]. PVDF, with repeated -(CH_2_CF_2_)_n_- units, manifests excellent chemical and mechanical resistance, combined with a good thermal stability, which makes it one of the most preferred materials for membrane fabrication. The use of semiconductor photocatalysts and their blends, such as TiO_2_ [25], TiO_2_/SiO_2_ [27,28], TiO_2_/RGO [29], TiO_2_/GO [17,18] and g-C_3_N_4_/RGO [19], in the fabrication of photoactive membranes, has attracted considerable interest in recent years. In addition, mainly due to the possibility of being excited by visible light, ideally by using solar light, some of these photocatalysts are able to degrade pollutants as well as to provide antifouling properties due to the presence of high densities of oxygen groups on the photocatalyst particles. Table 1 presents a list of studies related to photocatalytic processes for contaminants’ abatement in membrane technology.

Different procedures have been applied for introducing the photocatalyst either on the membrane surface or inside the matrix. These methods mainly include dip coating, photochemical method [32] layer-by-layer assembly [33,34], physical deposition [35], phase inversion [35], biaxial stretching [36], plasma reduction [37]. The immobilization of nanoparticles on the membrane top surface, although offering the possibility to improve photocatalyst performance due to the presence of active sites on the membrane surface, could result in the leaching of photocatalyst nanoparticles during operation [17,38]. In contrast, the preparation of microfiltration (MF)/ultrafiltration (UF) membranes by polymer-photocatalyst blending and subsequent casting of the suspension, although less employed, is more adequate for better anchoring the nanoparticles inside the polymeric matrix and for conferring anti-fouling properties to the membrane [39,40,41,42].

Photocatalysts of nanometric dimension are better choices compared to the traditional large particle size semiconductors [43]. Because of their superior properties, including noticeable thermal stability, high mechanical strength and huge surface interaction, carbon-based nanomaterials are extensively studied [44,45,46,47,48]. Graphene oxide (GO) offers the possibility of the covalent anchoring of organic compounds because of its abundant epoxy and hydroxyl groups located on the basal plane and carboxyl and carbonyl groups placed on the edges of the flat sheets [49,50]. In photocatalysis, GO has attracted great attention due to its easy synthesis, structure and the presence of various functional groups [51,52]. The photocatalytic activity of GO nanostructures was investigated by Krishnamoorthy et al. [53] for the photoreduction in resazurin into resorufin as a function of time under UV irradiation. Hou and Wang [54] reported the use of GO as metal-free photocatalyst producing millimolar levels of H_2_O_2_ under visible light irradiation. The photocatalytic activity of GO derives from its semiconductor behavior with an absorption band at about 280 nm, extending near the visible region. The activity of GO has been found to grow with its oxygen content.

In the present work, we reported the preparation of organic–inorganic polymeric membranes for environmental applications. The sustainability of the investigated techniques is given by the use of TEP as non-toxic solvents for the membrane preparation together with the visible light photocatalytic degradation of an organic dye for membrane applications.

GO was synthesized by modified Hummers method [51] while membranes were prepared via Non-Solvent Induced Phase Separation (NIPS) and Vapour-Induced Phase Separation (VIPS)/NIPS. 

The photocatalytic performance of PVDF-GO membranes was examined for the degradation of methylene blue (MB^+^) under visible light irradiation (λ > 420 nm). The effect of pH and the presence of sodium chloride in the dye aqueous solution were also investigated. 

## 2. Materials and Methods

### 2.1. Materials

PVDF (Solef 6010) was purchased from Solvay Specialty Polymers S.p.A. (Bollate, Italy), polyethylene glycol (PEG, *M*_w_ = 200 g·mol^−1^) from Sigma-Aldrich now Merck (Milan, Italy) and polyvinyl pyrrolidone (PVP, *M*_w_ = 9000 g·mol^−1^) from Basf (Mannheim, Germany) were used as pore former additives. Graphite (synthetic fine powder, particle size >20 microns from Sigma-Aldrich), H_2_SO_4_ (98%), KMnO_4_ (99%) and H_2_O_2_ (37% aqueous solution) were purchased from Sigma Aldrich. TEP and MB^+^ (*M*_w_ = 373.9 g·mol^−1^) were purchased from Sigma-Aldrich (Milan, Italy).

### 2.2. GO Preparation and Characterization

GO was prepared starting from graphite that was submitted to Hummers-Offeman oxidation followed by the exfoliation of resulting graphite oxide by sonication as described by Ruoff and coworkers [55]. Briefly, graphite powder (3 g) was dispersed in a mixture of concentrated H_2_SO_4_/H_3_PO_4_ (360:40 mL) in an ice bath at 0 °C. KMnO_4_ (18 g) was carefully added to this suspension in small portions. A strong exothermic reaction raising the temperature to 35–40 °C was observed. CAUTION: KMnO_4_ addition has explosion risk! After total addition of KMnO_4_, the reaction mixture was then heated to 50 °C under stirring for 12 h to complete oxidation. Then, the reaction was cooled to room temperature and poured into 400 g of ice containing 30% H_2_O_2_ (3 mL) to decompose the excess of KMnO_4_. After air-cooling, the suspension was filtered, washed with 1:10 HCl (37%) solution and then with water. The resulting graphite oxide obtained in the oxidation was sonicated using a 700 W horn in 400 mL of water for 30 min and centrifuged at 4000 rpm for 4 h. The supernatant was centrifuged at 15,000 rpm for 1 h. The supernatant resulting after centrifugation at 15,000 rpm was collected and dried at 60 °C to obtain GO.

The obtained GO was characterized in terms of ICP-optical emission spectroscopy (ICP-OES), transmission electron microscopy (TEM), atomic force microscopy (AFM), X-ray photoelectron spectroscopy (XPS) and infrared spectroscopy (FTIR), as described in the Appendix A.

### 2.3. Membrane Preparation and Characterization

Membranes were prepared using a combination of NIPS and VIPS methods [26,56]. PVDF casting solutions were prepared using TEP as a solvent with different weight ratios according to Table 2. Three different types of membrane were studied. Dope solution containing just the polymer and the solvent was cast at different exposure times (0, 2.5, 5 min) under controlled humid air (~55%) and temperature (25 °C) inside a climatic chamber (DeltaE S.r.l., Rende, Italy). The presence of additives, i.e., PVP and PEG, was also investigated. Finally, GO was incorporated in membranes exposed at different delayed times during the VIPS step. Table 2 summarizes the membranes produced by changing the operational conditions. For the preparation of photocatalytic membranes, first GO nanosheets were dispersed in TEP using an ultrasonic bath (USC600TH, VWR, Oud-Heverlee, Belgium) for 2 h, then the additives and polymers were added and the dope solution was stirred at 100 °C for 12 h to ensure the formation of a homogeneous clear solution. Next, the solution was degassed for 6 h before casting. All dope solutions were cast on a glass plate (20 cm × 30 cm) by using a manual film applicator set at 400 µm (Elcometer 3700/1 Doctor Blade, Aalen, Germany; adjustable gap size: 30–4000 μm). The membranes were washed with 60 °C bidistilled water three consecutive times and then kept in bidistilled water for 12 h to remove residual solvents; then, the membranes were dried in air for 12 h and finally heated in an oven for 4 h at 50 °C.

Membranes were characterized in terms of SEM, AFM, contact angle (CA), porosity, pore size, mechanical properties and pure water permeability (PWP) following the procedures described in the Appendix A.

### 2.4. Photocatalytic Degradation Test

Photocatalytic performance of the membranes was tested by using MB^+^ cationic dye as an organic contaminant with a molecular weight of 284 Da. Photocatalytic experiments were conducted by means of a cross flow cell with a quartz window on the top side, as shown in Figure 1. Before photocatalytic tests, in order to limit the phenomenon of adsorption, the membrane sample was conditioned overnight in a 50 mL dye solution (10 µmol/L). Photocatalytic experiments were performed by circulating 150 mL of a 10 µmol/L MB^+^ solution in both darkness and under visible light irradiation. Flow velocity and trans membrane pressure (TMP) were fixed at 1.5 L/h and 0.2 MPa, respectively. The distance between a 150 W xenon lamp (E40 4300 k 220–240 V 50–60 Hz) and cross flow cell was adjusted at 13 cm. A cut-off UV irradiation filter (GG420 Colored Glass Filter, 420 nm long pass with 25 mm diameter FGL420, Thorlabs, Newton, NJ, USA) was employed to filtrate UV light wavelength. The wavelength range was between 400 and 1800 nm. Irradiation was started after a preliminary stabilization time of the system (after 45 min recirculation of the solution) and, every 30 min, 1 mL of solution was taken from the feed tank. The concentration of MB^+^ was measured using a UV/VIS spectrophotometer LAMBDA EZ 201, Perkin Elmer (city, country) at 664 nm. The MB^+^ solution was continuously run through the system so that the retentate and permeate were recirculated back inside the feed tank.

## 3. Results

### 3.1. Characterization of the GO Nanosheets

GO was obtained by the deep chemical oxidation of graphite and exfoliation by sonication, as previously reported [55]. The resulting GO dispersion was centrifuged up to 15,000 rpm to remove those sediments that should correspond to incompletely exfoliated material. Only the supernatant resulting after high-speed centrifugation, that must be constituted by preferentially single-layer GO sheets of lateral size from 0.8 to 3.0 microns, was used in the present study. This was confirmed by AFM measurements (Figure 2a,b) of the GO suspended in aqueous medium. Analysis of a statistically relevant number of GO particles showed that most of them have a sheet thickness of 1.0 nm, in the range of the expected values for hydrated single-layer GO [55]. TEM images showed wide fields full of GO nanosheets exhibiting a typical bidimensional morphology and lateral dimensions in the micrometer range with light contrast due to its single-layer configuration (Figure 2c).

GO was characterized by combustion chemical analysis, measuring a percentage of carbon slightly above 50%. The presence of sulphur, probably associated with the presence of sulfonic and sulfate groups bonded to the GO sheets, was also determined in this analysis. Metal impurities, mainly manganese, were also detected by ICP-OES. Manganese was probably present as Mn^2+^ due to the chemical reduction in MnO_4_^−^ by H_2_O_2_ and associated to carboxylate groups of GO. Appendix A reports the Mn content of the samples under study. As already reported, IR spectroscopy of GO shows the presence of oxygenated functional groups, mainly carboxylic acids, hydroxyl and epoxy groups. The presence of oxygenated groups was also determined by XPS analyses (Figure 2d). It should be noted that, although XPS analysis can be carried out on GO powders, in the present case, XPS analysis was performed on thin films after depositing a drop of the GO aqueous suspension on quartz and allowing water to evaporate, before introducing the sample in the XPS machine. Both survey and high-resolution XP spectra showed an atomic carbon:oxygen ratio for the exposed GO surface of 1.7 based on the relative peak areas after correction by the response factor of each element (Appendix A). Deconvolution of the broad C1s peak in XPS fitted to four components corresponding to sp^2^ carbons (284.5 eV, 22%), carbon atoms bonded to oxygen with single (286 eV, 18%) and double bond (287 eV, 27%) and carboxylic acid groups (289.0 eV, 33%). Figure 2d shows the experimental high-resolution XPS C1s peak and its best deconvolution to the individual components. As shown in Raman spectrum (see Appendix A), in contrast to ideal graphene, where the shape and position of the 2G peak can be used to determine the number of layers, the broadness of the Raman peak at 2900 cm^−1^ in the case of rGO encompassing the G + D and the 2D bands does not allow analogous analysis. In the case of GO, sub-nanometric AFM measurements are the safest technique to assess the single-layer nature of the GO sample used in the study. Overall, all the previous data are in agreement with those expected for a single-layer GO previously reported in the literature [55].

### 3.2. Membrane Characterization

SEM images of M1–M3 membranes, prepared with the polymer and the solvent at different exposure times, are presented in Figure 3. 

SEM images evidenced a compact structure for M1, M2 and M3 membranes with no relevant differences between them. A dense layer on the top surface and a spherulitic matrix was formed for the three membranes. This observation could be explained by considering the high hydrophobicity of PVDF, which caused a fast demixing with the formation of a skin layer [57].

Figure 4 and Figure 5 show SEM pictures of M4–M6 and M7–M9, respectively, in which both types of membranes were prepared by including PVP and PEG in the casting solution formulation.

The influence of the NIPS/VIPS preparation procedure, during which the nascent films were exposed for different times to water vapour, was observed for both membrane types prepared with and without GO.

The solvent–non-solvent exchange rate represents one of the most affecting parameters during membrane formation. It is well known, for example, that the presence of water in vapour form during VIPS step, promotes a delayed demixing, thus favoring the formation of a more porous, sponge-like structure in comparison with NIPS process. However, it should be mentioned that, during polymer precipitation, kinetic factors also play a relevant role for the polymeric chains’ arrangement. Thermodynamic factors are related to polymer precipitation behavior, while kinetic factors are related to the exchange rate of solvent/nonsolvent during phase separation [58]. The combination of the two factors results in different membrane structure and properties [59]. 

As can be seen in Figure 4, M4 membrane was significantly affected by the addition of pore-forming additives compared to the analogue M1 membrane formed from a PVDF-TEP solution. The M1 membrane showed a high compact structure, while M4 exhibited a finger-like structure on the top surface, surrounded by a cellular, porous layer. The presence of macrovoids across the entire cross section was observed for M7, which differed from M4 for the presence of GO.

The addition of additives dramatically increased the miscibility of solvent with non-solvent, therefore the creation of fingers and macrovoids was favored for M4 and M7. When the NIPS/VIPS method was adopted, the demixing rate between the solvent and the non-solvent was delayed, thus the presence of fingers and macrovoids gradually disappeared, with the formation of completely spongy membranes when the time interval reached 5 min [60]. The addition of PVP and PEG as pore-forming agents and GO nanosheets into the dope solutions increased the hydrophilicity of the dope solution. Due to its high solubility in water, PEG instantaneously dissolved in water during NIPS and left large macrovoids inside the membrane [61]. This analogy can be clearly observed in cross-section images of M1, M4, and M7 reported in Figure 3, Figure 4 and Figure 5, respectively. The presence of PVP and PEG as pore-forming agents, together with GO, led to different morphologies, depending on the exposure time of the nascent film to water in vapour form. On the one hand, a porous matrix with macrovoids was observed when immersion-precipitation was adopted; in this case, the hydrophilic character of both additives and catalyst promoted a rapid TEP-water exchange. On the other hand, the increase in polymeric solution viscosity, due to the presence of PVP, PEG and GO as opposed to the mass transfer during phase inversion, led to a spongy matrix when the NIPS/VIPS procedure was followed and the nascent film was exposed to humid air for 5 min (Figure 5).

The exposure time to humidity (from 2.5 to 5 min) greatly affected the morphology of the membranes prepared with additives (Figure 4 and Figure 5). The exposure of the cast film to a humid environment promoted the formation of a more porous structure on the surface while, along the cross-section, it favored the formation a spongy architecture. The porogen effect of humidity is a quite well studied phenomenon and it is related to the adsorption, by the cast film, of water molecules from the humid air, which delays the phase-inversion process, fostering the creation of a porous and sponge-like structure [25,62,63,64,65].

Figure 6 presents the morphology of M10, i.e., the analogue M8 membrane (0.125 wt % GO) prepared by increasing GO content (0.5 wt %). The increase in GO concentration in the casting solution favored the increase in solution viscosity, thus delaying the demixing rate with the formation of a symmetric, sponge-like structure. The presence of GO nanosheets in the M8 and M10 can be clearly observed in the images provided in Figure 7 taken at a higher magnification.

The effects of modification on the top surface morphology and roughness parameter of the fabricated membranes were also studied by AFM (Figure 8). The topography was measured on five different areas of the membrane surface and the root mean square (RMS) roughness (Sq), roughness average (Sa), and peak to peak value (Sz) were calculated. The obtained data and the respectively standard deviation are reported in Table 3.

Based on the extrapolated data and the series of images, it is possible to notice the following features:For membranes prepared with only polymer and solvent (M1, M2 and M3 membranes), the roughness decreased with increasing exposure time to humidity (0–2.5–5 min). This result is in agreement with the SEM images (Figure 4) that show a dense skin layer on the top surface;For membranes obtained using PVP and PEG (M4, M5 and M6 membranes) as pore-forming agents, the roughness increased in relation to the humidity exposure time. In fact, these three membranes presented a more porous morphology, as already observed in the SEM images (Figure 5). In particular, the M6 membrane presented the highest roughness value;With respect to M4, M5 and M6 membranes, when GO nanosheets at 0.125 wt % were added to the dope solution (M7, M8 and M9 membranes), roughness value remained constant. The only exception was at the highest humidity exposure time of 5 min (M6 and M9 membranes), where the roughness (Sa) exhibited a decline from 70.12 nm (M6) to 32.82 nm (M9);Moreover, when the 0.5 wt % of GO was used (M10 membrane), a membrane with higher roughness was observed. This result could be explained by the agglomeration of particles in the mixed-matrix membrane and the loss of uniformity due to the formation of more pores on the surface arising from the lower miscibility of the dope solution [66];In all cases, for all the membranes prepared with PVP and PEG, the average roughness was enhanced compared to M1–M3 membranes. Furthermore, M7–M9 membranes with higher cavities and pores on the surface possessed higher roughness that led to an increased permeability. The intensification of GO concentration in membranes generally exhibited higher roughness compared to those with lower concentrations (as in the case of M10).

For water treatment applications, membrane hydrophilicity, assuring water passage through membrane pores and reducing fouling, represents a key parameter.

Typically, to measure the hydrophilic properties of the membranes, CA is used. The values of CA for the membranes (top side and bottom side) are listed in Table 4. According to CA measurements (top sides), the hydrophilic character of M1–M3 membranes was independent from delay time as well as SEM, AFM and PWP characterization, and may be due to PVDF chains arrangement during phase inversion. The presence of hydrophilic pore-forming agents in the dope solution, i.e., PVP and PEG, led to the reduction in membrane CA for M4 (CA 65 ± 2°) prepared via NIPS. However, by coupling NIPS with VIPS, more hydrophobic membranes were obtained, and the CA increased with the increase in exposure time to a controlled relative humidity from 2.5 to 5 min. In fact, M5 showed a CA of 77 ± 2°, while M6 had a CA of 108 ± 3°. These results may be related to the membrane roughness, which increased concomitantly by extending the interval times during the VIPS step. Similar results were reported in the literature [67]. Decreased roughness means smoother surfaces for M7–M9 membranes, which in turn results in a decreased contact angle [68]. The increase in membrane hydrophilicity was due to the well-distributed GO nanosheets on the top surface of the membranes that attracted water molecules and created a thin shielding water layer on the pores [69]. This was at the basis of the very hydrophilic nature of M10 membrane prepared with the highest concentration of GO.

CA assessment for M7–M9 membranes revealed a positive effect of GO on membrane hydrophilicity. In fact, the incorporation of GO nanosheets in the polymeric matrix promoted the formation of oxygen functional groups (such as hydroxyl and carboxyl groups) on the top surface, increasing the hydrophilic membrane character. However, similarly to what has been observed for MB-type membranes, when the exposure time to water, in vapour form, passed from 0 to 2.5, until 5 min, the CA increased, due to the increase in membrane roughness [70]. Representative CA pictures of M1 and M7 membranes are reported in the Appendix A.

The results obtained for the membrane pore size and PWP are reported in Figure 9. By increasing the time intervals during the VIPS step, M1–M3 membranes, having a pore diameter in the range between ~0.06 and ~0.08 m, were not significantly affected; in other words, the exposure time to relative humidity did not play a relevant role for membrane pore size determination. On the contrary, by varying the casting solution composition, evident changes in mean flow pore diameter were detected.

The addition of PVP and PEG, as shown and explained in SEM pictures, fostered the formation of a more porous structure, which resulted in membranes with a larger pore size and, therefore, a higher PWP.

M4 showed a pore size of ~0.1 m, increased up to ~0.5 m for M5 and reached a value of ~0.7 m for M6. Hence, the combination of a PVDF-TEP and PVP-PEG dope system, together with VIPS/NIPS preparation procedure, allowed for tailoring the membrane pore size. By adding GO to the dope solution, a decrease in the membrane pore size for M8 and M9 was noted. Although further investigations should be necessary to better understand the reduction in membrane pore size, a first explanation could be associated with the dope solution viscosity, which increased when GO nanosheets were added to the polymer-solvent-additives solution. Similar results are reported in the literature by Zhu et al. [71]. Furthermore, the competition between the solvent and the nano-fillers to exchange with the non-solvent caused a lower mean flow diameter. By using a higher GO concentration, i.e., 0.5 wt %, the resulting M10 membrane exhibited a pore size in the MF range (~0.35 m). The PWP results are presented in Figure 9. M1–M3 membranes showed no PWP up to three bars. The reason for this can be ascribed to a compaction phenomenon which may have occurred during membrane pressurization, which caused a constriction of the small pores, blocking the permeation of water. The analogue M4 and M7 membranes, on the contrary, resulting from solutions containing additives and GO-fillers, respectively, allowed water to pass across membrane pores. Specifically, M4 exhibited a PWP of ~150 L·m^−2^ h^−1^ bar^−1^, while M7 had a ~110 L·m^−2^ h^−1^ bar^−1^. For the membranes prepared via NIPS/VIPS, the PWP reflected the pore size results: PWP increased when the exposure time to non-solvent in vapour form passed from 2.5 to 5 min. In general, the increase in exposure time to humidity promoted the formation of membranes with a larger pore size and higher PWP.

The best results in terms of PWP were obtained for M6 (~3400 L·m^−2^ h^−1^ bar^−1^), which had a mean flow pore diameter of ~0.7 m.

Membrane porosity is strictly correlated with the membranes’ morphology. All the membranes prepared with the casting solution composed of PVDF and TEP (M1–M3), exhibited a porosity near to or lower than ~80%, which is in accordance with the SEM images shown in Figure 3. On the contrary, by adding PVP and PEG to the dope solution, the resulting membrane presented a higher porosity, which further increased with the presence of GO, reaching a value of ~87%. However, a GO content of 0.5 wt % caused a reduction in porosity for M10, probably due to the blocking of pores by catalyst nanosheets’ aggregation. The results of the tensile strength and elongation tests are shown in Table 5. The Young’s modulus for all of the membranes was improved when GO, together with PVP and PEG, was added to the dope solution in comparison to the analogue polymeric pristine PVDF membranes. These results could be explained in terms of GO’s superior mechanical stability, high surface area and aspect ratio [40]. The positive effect of GO on the membrane mechanical features is confirmed by the elastic modulus of M10, for which a value of ~45 N/mm^2^ was detected. Elongation at break revealed high values for all the prepared membranes, with a maximum of ~50% for M8. However, for M9 and M10, a decline in membrane elongation was observed, probably due to GO aggregation at the point of breakage.

Carbon derivatives, due to their excellent mechanical properties, are the best additives for improving the mechanical strength of the membranes. A large number of functional groups on the GO nanosheets’ surface can create a stable cross-linked network with the PVDF polymer. Hence, GO nanosheets can increase the mechanical properties of the membrane due to more favorable inherent properties.

### 3.3. Photocatalytic Tests

In order to evaluate the photocatalytic activity of M7–M10 membranes containing GO, the degradation of MB^+^ under visible light was performed. Three different phenomena, which can occur during the experimental tests, can be responsible for the dye concentration decrease in the feed solution: the spontaneous degradation of MB^+^ by light, adsorption of MB^+^ by the membrane and effective degradation of MB^+^ through the photocatalytic activity of GO.

In order to consider exclusively the dye degradation related to the catalyst activity, the impact of the first two phenomena (degradation by light and adsorption) was also evaluated. Dye solution degradation tests, in the absence of a membrane, were conducted by exposing the dye solution to visible light. A degradation of about 20% was observed under these conditions and it was considered as a base line (Figure 10). The dye removal related to an adsorption mechanism, and therefore not connected to the catalytic activity of GO, was considered by conditioning the membrane overnight in a 50 mL dye solution (10 µmol/L) before each test in order to reach an adsorption equilibrium. Moreover, photocatalytic tests with M4–M6 membranes (catalyst free) were performed under light irradiation, resulting in a removal of the dye of up to 52.6% (in case of M5), as shown in Figure 10 (this removal was exclusively due to adsorption mechanism and MB^+^ spontaneous degradation under light irradiation). Tests carried out with M4–M6 membranes (catalysts free under visible light) allowed to discriminate the photocatalytic activity of the catalysts from other phenomena occurring during the tests.

As can be seen in Figure 10, all the membranes prepared with GO (M7–M9) presented higher MB^+^ degradation efficiency in comparison to the others as a consequence of the photocatalytic contribute of GO.

In particular, the photocatalytic M8 membrane showed the maximum degradation value of 83.3%, followed by M7 (about 68%) and M9 (about 62%). The three membranes presented the same GO content (0.125 wt %), and they only differed in their exposure to a humid environment (from 0 to 5 min) during their preparation. The reason for their different catalytic behavior might be, therefore, related to their different membrane morphology. M7 presented a more compact and denser structure, as evidenced by SEM and by the lower PWP values, resulting in a decreased contact surface with the dye solution. A more spongy-like structure (like the case of M8 and M9 membranes) can offer, in fact, more contact sites between the catalyst and the dye, favoring its degradation. However, M9 membrane, which was the more porous and permeable membrane, exhibited the lowest degradation performance. This could be related to the formation of a too-loose structure, which did not favor the contact between the catalyst and the dye solution.

The effect of membrane morphology on dye degradation can also be observed in the results reported in Figure 11. In this graph, the removal efficiency of the membranes M5 (0 wt % GO), M8 (0.125 wt % GO) and M10 (0.5 wt % GO), differing in formulation just in the GO content, is displayed. As can be seen from the graph, despite the higher concentration of GO, M10 membrane did not show the highest degradation rate. This could be due to its different morphology and pore size. In this case, in fact, the presence of GO in the polymer solution favored the formation of a membrane with higher pore size, as shown in Figure 9, which negatively affected the MB removal efficiency due to a lower contact of the membrane catalytic sites with the dye solution.

The degradation of MB using GO, often anchored with various metal oxides, has been reported by many authors in the literature [72,73]. The MB^+^ is degraded following a pathway which culminates with the total conversion of the dye in water, carbon dioxide and inorganic anions (SO_4_^2−^; NH_4_^+^ and NO_3_^−^). Most of the time, the formation of a series of intermediates occurs [74,75].

### 3.4. Influence of pH on Removal Operation

Since GO contains a large percentage of oxygen groups such as carboxylic acid (COOH), ketone (C=O), hydroxide (OH) and epoxy (C–O), pH plays a strong influence on its charge, therefore influencing its adsorption capacity by electrostatic interaction. The great adsorption capacity of GO is attributable to the negative charge of COO^−^ groups and π–π interaction with aromatic groups of adsorbent molecules. The zeta potential charge of the membrane, which is dependent on the pH of the solution, was measured by Malvern Zetasizer, exhibiting a change from positive to negative charges in GO at pH values around 4. This is compatible with the protonation/deprotonation of the carboxylic acid groups present on GO as the main factor responsible for the charge present on the nanosheets.

Due to these changes in Coulombic charges and adsorption, the photocatalytic activity of the membranes is also strongly influenced by the solution pH. Figure 12 displays the photodegradation of an MB^+^ solution for the M8 membrane at different pH values (3, 7 and 9), adjusted using 0.2 M NaOH and HCl. As can be seen, MB^+^ degradation decreased in the acidic and basic conditions, reaching its maximum under a neutral pH.

Degradation efficiency in acidic solution was significantly reduced, which could be rationalized by the occupation of active sites on the membrane by protons, resulting in a change in the Coulombic charge of the GO nanosheets and a change in its adsorption properties. Membrane surface charge below the point of zero charge (pH_pzc_~6) is positive, and above that, value is negative and, as mentioned before, MB^+^ is a cationic dye because of the amine group. Therefore, for the dye solution with pH values of less than pH_pzc_, electrostatic repulsion resulted in a dramatic decrease in degradation efficiency.

At quasi-neutral pH values, the formation of a complex between basic groups of GO (beyond pH 6) and cationic dye molecules by electrostatic attraction should favor adsorption on the GO nanosheets and, therefore, also promote photocatalytic degradation. A further increase in pH to strongly basic values should again disfavor the adsorption of MB^+^ on GO by competition with the high concentration of counter cations present in the solution, resulting again in a lower photocatalytic activity.

### 3.5. Salt Effect Assessment in Dye Removal Efficiency

In order to investigate the influence of salt presence on the removal process, dye solutions containing 0.001 and 0.01 mol/L of NaCl were tested with the M8 membrane and the results are shown in Figure 13. As can be seen from the graph, the presence of NaCl caused a decrease in membrane removal efficiency, which was particularly evidenced at a high salt content (0.01 M). Chloride ions in the solution formed a stable complex with MB^+^ and, therefore, MB^+^ was involved in a competition between adsorption on the membrane and pairing with chloride ions [76]. The competition of the dye for a complex formation at stronger ion strengths (higher salt concentrations) was increased, which resulted in a lower availability of the dye for the photocatalytic membrane sites and, therefore, a lesser degradation efficiency. On the other hand, as a consequence of this competition, the number of MB^+^ molecules adsorbed on the membrane surface was decreased and the photocatalytic capacity of the membrane was compromised. The decrease in process efficiency as a consequence of Cl^−^ anions presence is in agreement with the literature data [77,78]. The decrease in removal efficiency might be, in fact, related to Cl^−^ ions in the dye solution which decreased the electrostatic attraction between MB^+^ and the active sites of the membrane (screening effect) [79].

## 4. Conclusions

In this work, four types of membranes were prepared by using a non-toxic solvent and, for the first time, GO nanosheets were used as a metal-free catalyst to photoactivate the membrane under simulated solar light (visible light) irradiation. The influence of exposure time (0, 2.5 and 5 min) of the membrane to a humid environment before coagulation in a non-solvent bath was investigated for three sets of membranes prepared with polymer and solvent, with the addition of pore-forming additives and with the addition of the photocatalyst GO. The concentration of GO in the membrane was also varied. The membrane structure changed from a dense and compact morphology in the absence of PVP and PEG as additives to a finger-like architecture in presence of the pore formers. In the latter case, the increase in membrane exposure time to humidity enhanced the formation of a sponge-like structure. This structure was further promoted by the addition of GO. The addition of pore-forming agents and GO also led to an increase in the roughness, in particular for higher GO concentration.

The incorporation of GO significantly improved the mechanical strength of the obtained membranes as well as their wettability, due to the large number of hydroxyl groups on the layer surface. The development of a hydration layer on the membrane surface can decrease the attachment of pollutants and microorganisms, protein and pharmaceutical foulants, making the membrane less prone to fouling onset.

M8, prepared with PVP, PEG and GO (0.125 wt %) in the dope solution, and exposed for 2.5 min to moisture during the VIPS step, exhibited good photocatalytic performance towards MB^+^ dye, used as a model pollutant under simulated solar light irradiation, reaching a dye removal efficiency of 83.5%.

Photocatalytic experiments with pH values of 3, 7 and 9 were performed, showing that quasi-neutral pH is an optimum value for the photocatalytic process. The presence of salt (NaCl) negatively affected the photocatalytic process, causing a decrease in the dye degradation efficiency, probably related to Cl^−^ ions in the dye solution which decreased the electrostatic attraction between MB^+^ and the active sites of the membrane.

Overall, the present results show that by using environmentally greener solvents and procedures, it is possible to prepare functional PVDF membranes that exhibit good photocatalytic activity employing GO as a catalyst. This functionality should allow for the development of more efficient membranes using less toxic solvents.

## Figures and Tables

**Figure 1 polymers-12-01509-f001:**
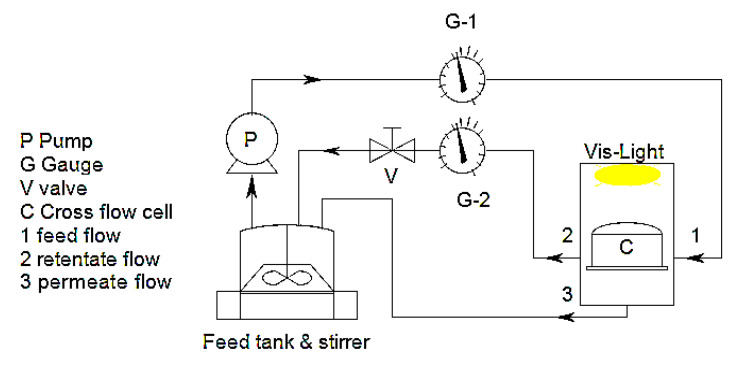
Schematic of photocatalytic cross flow system.

**Figure 2 polymers-12-01509-f002:**
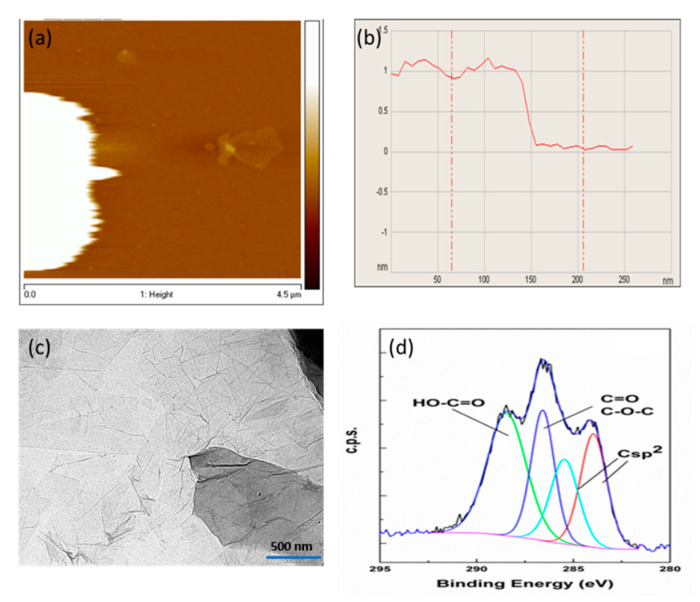
Exfoliated graphene oxide (GO) monolayer atomic force microscopy (AFM) image by non-contact mode (**a**) and vertical height of the sheet (**b**); (**c**) TEM image (scale bar 500 nm); (**d**) Experimental, high-resolution XPS C1s peak of the GO employed in the present study and its best deconvolution fit to individual components.

**Figure 3 polymers-12-01509-f003:**
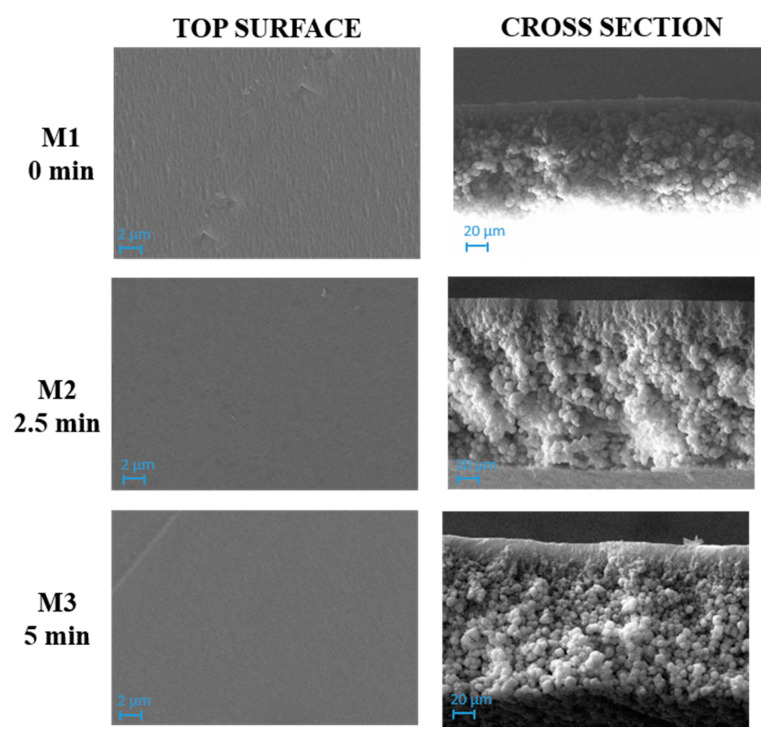
SEM images of M1, M2 and M3 membranes prepared with polyvinylidene fluoride (PVDF) and triethyl phosphate (TEP) and exposed for 0, 2.5 and 5 min to humidity, respectively. Magnification: top surface 10,000×, cross section 1000×.

**Figure 4 polymers-12-01509-f004:**
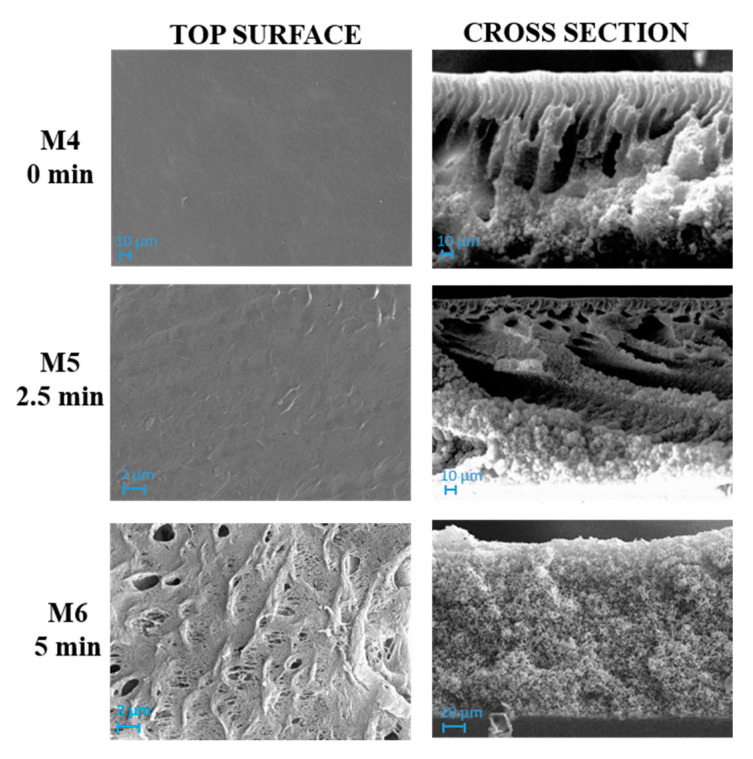
SEM images of M4, M5 and M6 membranes prepared with PVDF, PVP, PEG and TEP and exposed to humidity for 0, 2.5 and 5 min, respectively. Magnification: top surface 10,000×, cross section 1000×.

**Figure 5 polymers-12-01509-f005:**
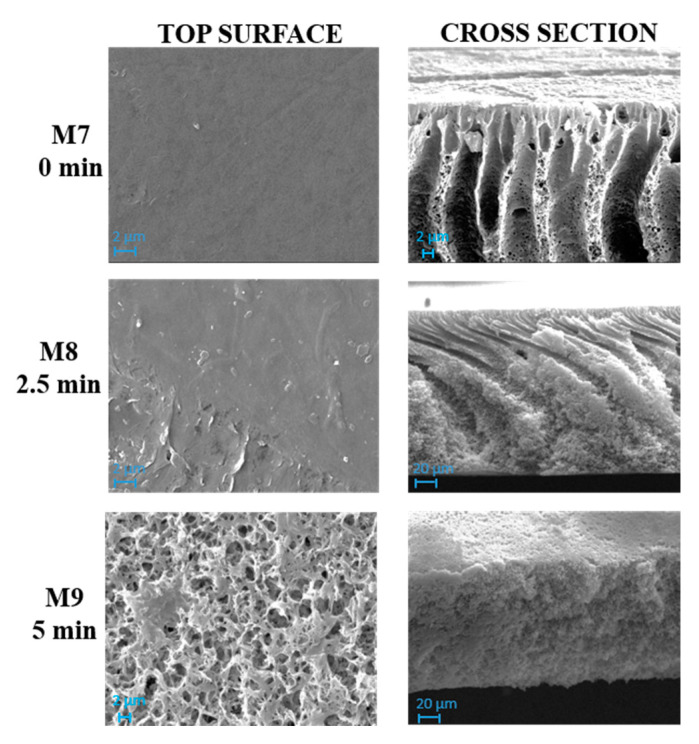
SEM images of M7, M8 and M9 membranes prepared with PVDF, PVP, PEG, GO and TEP and exposed to humidity for 0, 2.5 and 5 min, respectively. Magnification: top surface 10,000×, cross section 1000×.

**Figure 6 polymers-12-01509-f006:**
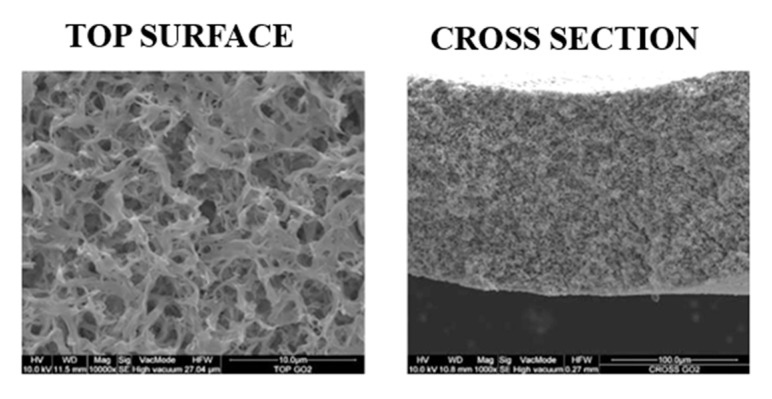
M10 SEM images. Magnification: top surface 10,000×, cross section 1000×.

**Figure 7 polymers-12-01509-f007:**
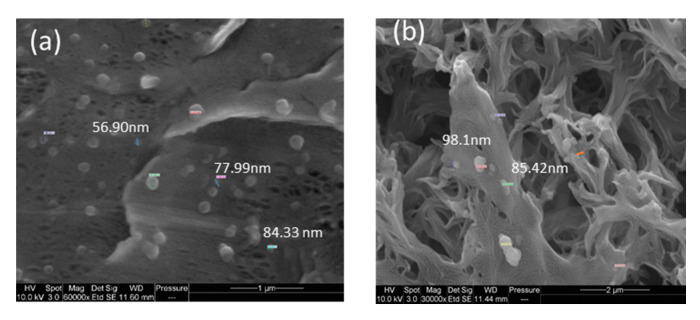
Images showing the presence of GO nanosheets: (**a**) M8 top surface (Magnification: 60,000×) and (**b**) M10 top surface (Magnification: 30,000×).

**Figure 8 polymers-12-01509-f008:**
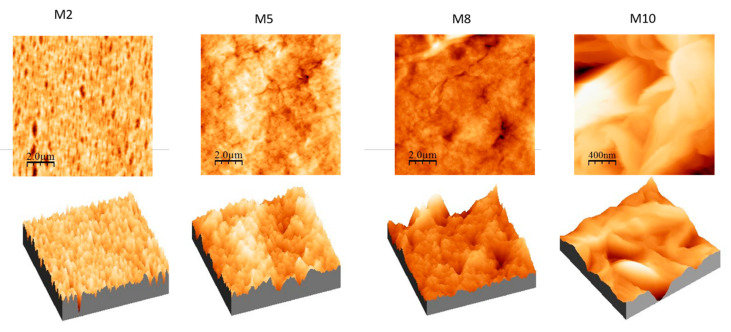
AFM images with corresponding 3D views of M2, M5, M8 and M10 membranes deriving from films exposed to humidity for 2.5 min during VIPS-NIPS.

**Figure 9 polymers-12-01509-f009:**
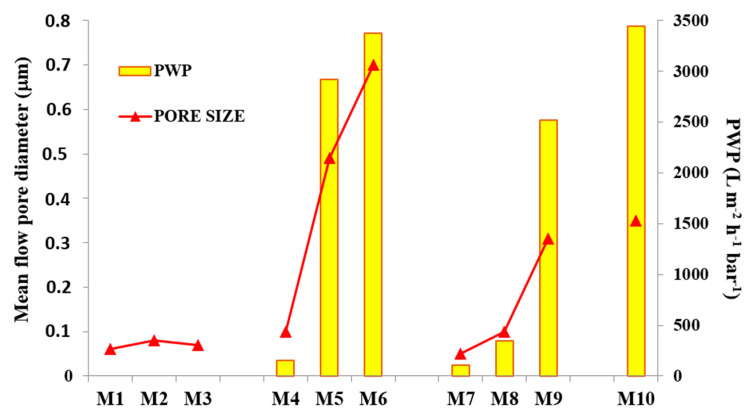
Pure water permeability (PWP) correlated with pore size diameter. Standard deviation was less than 5% in all operative conditions.

**Figure 10 polymers-12-01509-f010:**
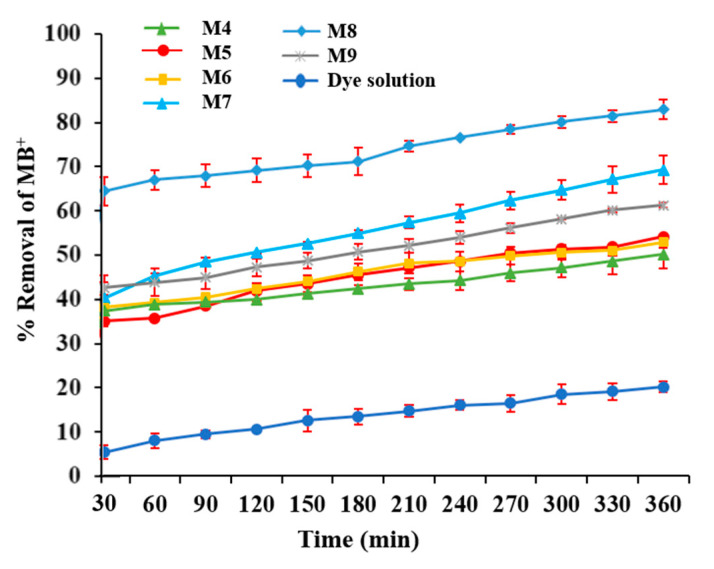
Comparison of dye removal using a catalyst-free (M4–M6) and GO-filled (M7–M9) membranes and dye solution degradation without a membrane under visible light irradiation.

**Figure 11 polymers-12-01509-f011:**
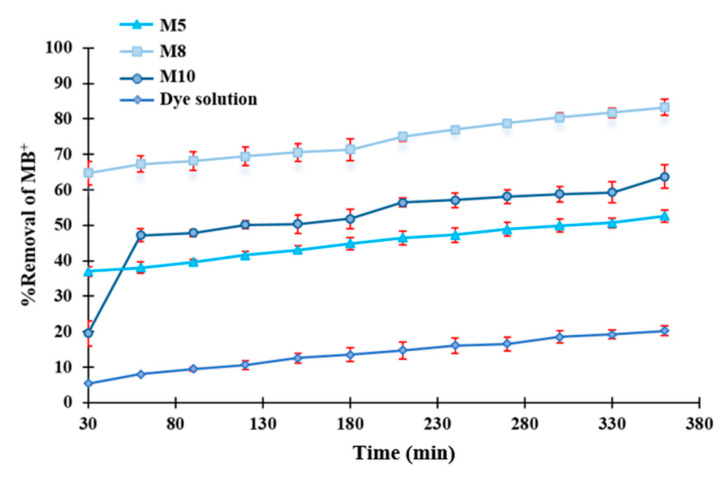
Dye removal using M5, M8 and M10 membranes and dye solution degradation without membrane under visible light irradiation.

**Figure 12 polymers-12-01509-f012:**
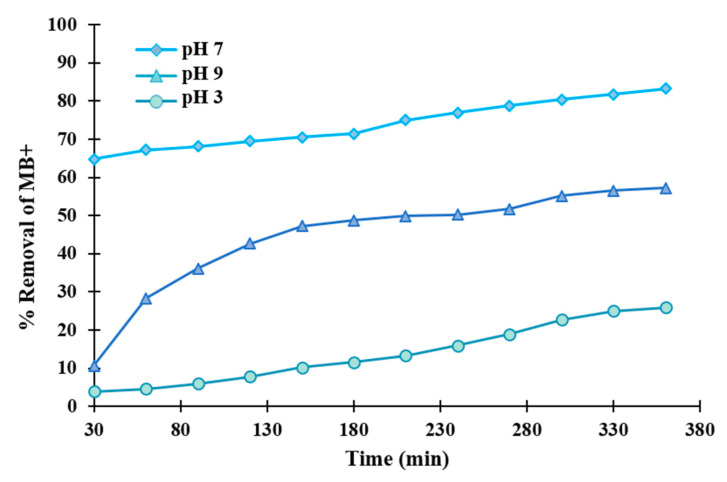
Influence of the pH of the dye solution on removal efficiency in M8 membrane.

**Figure 13 polymers-12-01509-f013:**
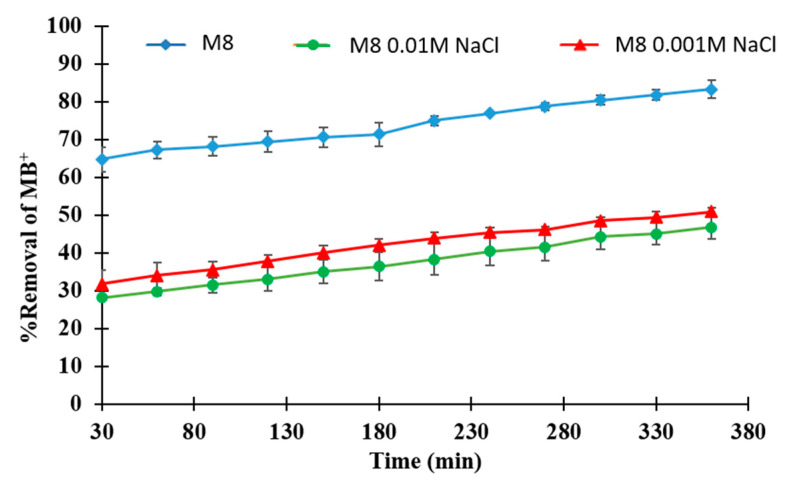
Effect of salty dye solutions (0.001 and 0.01 M NaCl) on the removal process for M8 membrane.

**Table 1 polymers-12-01509-t001:** Membranes containing photocatalysts for pollutants’ degradation.

Membrane	Photocatalyst	Immobilization Method	Irradiation Wavelength	Pollutant Removal Studied	Reference
**Ceramic**	GO and TiO_2_	Dip-coating technique	UV and visible light	Methyl orange and MB^+^	[29]
**Polysulfone**	GO and TiO_2_	Layer-by-layer assembly	UV and Sunlight	MB^+^	[18]
**Ceramic**	GO and TiO_2_	Dip coating method	UV and visible light	Azo dye pollutants	[8]
**Cellulose**	TiO_2_ and GO&TiO_2_	Assembling the nano particle on the flat sheet membrane	UV and visible light	diphenhydramine, methyl orange	[6]
**Fe_2_O_3_/TiO_2_/GO**	Fe_2_O_3_/TiO_2_/GO composite	Composite blending in inorganic membrane	Solar irradiation	Humic acid	[30]
**PVDF**	GO and TiO_2_	Blending solutions	UV light	Bovine serum albumin	[17]
**Cellulose acetate**	g-C_3_N_4_ NS and RGO	Assembly of nano composite on the commercial membrane	Visible light	Rhodamine B	[19]
**Polysulfone**	N-TiO_2_ and GO	grafted onto the membrane surface using a pump filter	UV and Sunlight	MB^+^	[31]

**Table 2 polymers-12-01509-t002:** Dope solution components of the prepared membranes.

Membrane CODE	PVDF (wt %)	PVP_K17_ (wt %)	PEG_200_ (wt %)	TEP (wt %)	GO (wt %)	Exposure Time to Fixed Humidity and Temperature (min)
M1	13	0	0	87	0	0
M2	13	0	0	87	0	2.5
M3	13	0	0	87	0	5
M4	13	3	24	60	0	0
M5	13	3	24	60	0	2.5
M6	13	3	24	60	0	5
M7	13	3	24	59.875	0.125	0
M8	13	3	24	59.875	0.125	2.5
M9	13	3	24	59.875	0.125	5
M10	13	3	24	59.5	0.5	2.5

**Table 3 polymers-12-01509-t003:** The roughness parameters of the membranes under study.

Membrane	Root Mean Square	Average Roughness	Mean Difference between Peaks and Valleys
	Sq (nm)	Sa (nm)	Sz (nm)
M1	21.01 ± 0.21	16.25 ± 0.10	165.09 ± 25.94
M2	17.76 ± 0.89	13.23 ± 0.62	179.30 ± 29.21
M3	11.89 ± 0.22	9.39 ± 0.21	125.50 ± 17.73
M4	33.86 ± 5.58	26.33 ± 3.82	271.66 ± 37.86
M5	33.25 ± 1.53	26.30 ± 1.11	281.84 ± 44.83
M6	91.90 ± 2.47	70.12 ± 0.84	753.38 ± 20.56
M7	34.10 ± 6.33	25.91 ± 4.06	224.01 ± 30.01
M8	30.61 ± 4.37	25.04 ± 3.22	186.19 ± 4.00
M9	43.41 ± 15.69	32.82 ± 11.81	267.75 ± 3.39
M10	194 ± 14.35	146 ± 10.15	1465 ± 30.80

**Table 4 polymers-12-01509-t004:** Thickness and contact angle (CA) of the prepared membranes.

Membrane	Thickness (µm)	CA	
		Top Side (°)	Bottom Side (°)
M1	121.4 ± 1.7	72 ± 3	102 ± 2
M2	123.7 ± 1.3	71 ± 3	106 ± 3
M3	132.5 ± 1.5	75 ± 1	112 ± 2
M4	144.1 ± 3.2	65 ± 2	106 ± 1
M5	159.0 ± 5.5	77 ± 2	117 ± 2
M6	170.5 ± 7.0	108 ± 3	130 ± 2
M7	158.6 ± 1.2	61 ± 2	100 ± 3
M8	161.0 ± 0.9	68 ± 2	129 ± 1
M9	163.4 ± 1.4	82 ± 3	125 ± 3
M10	162.5 ± 1.2	63 ± 3	118 ± 3

**Table 5 polymers-12-01509-t005:** Mechanical properties and porosity of the prepared membranes.

Membrane	Young’s Modulus	Elongation at Break	Porosity
	N/mm^2^	E%	%
M1	25.3 ± 0.3	27.2 ± 3.6	77.8 ± 0.5
M2	21.5 ± 0.5	20.4 ± 4.6	78.4 ± 0.4
M3	29.6 ± 0.1	40.5 ± 1.3	79.5 ± 0.9
M4	18.7 ± 3.4	40.9 ± 2.8	83.5 ± 0.8
M5	18.7 ± 3.3	22.5 ± 1.3	84.5 ± 0.4
M6	14.4 ± 2.2	47.0 ± 1.9	82.8 ± 0.2
M7	22.6 ± 3.4	30.8 ± 1.1	86.7.± 0.4
M8	22.1 ± 2.5	50.3 ± 2.4	87.5 ± 0.9
M9	23.2 ± 3.9	3.0 ± 1.2	86.4 ± 0.4
M10	44.4 ± 1.4	2.4 ± 1.3	83.7 ± 0.7

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
