# Peer review of "Polyvinylidene Fluoride-Graphene Oxide Membranes for Dye Removal under Visible Light Irradiation"

_polymers, 2020, doi:10.3390/polym12071509_

Round 1

Author Response

REVIEWER

  1. The description of the material and method is ambiguous. For instance, for graphene material synthesis the size of precursor is very important. However, the authors merely mentioned “fine particles” and doesn’t provide detailed information about graphite.

AUTHORS:

The supplier and some properties of the commercial synthetic graphite used as precursor of the current graphene membrane are now provided in the text (section 2.1) and in supporting information. It should, however, be commented that over the years many other graphite powders have been used to obtain graphite oxide with identical results. This point has also been commented in the revised version (line 169-170).

REVIEWER:

  1. Authors should include Raman spectroscopy of their synthesized materials as well as its size distribution. This would be important since they used very high centrifugation rpm 15000 to remove exfoliated from unexfoliated graphite, and I assume they should get pretty small definitely submicron particles.

AUTHORS:

As requested, the revised version now includes the Raman spectrum of GO (Figure S2 of supporting information). The lateral dimensions of the exfoliated GO sheets is between 0.8 and 3 microns. The referee is correct that this optimized separation protocol by centrifugation renders almost exclusively single layer GO. This data has also been indicated in the revision (line 174-176).

REVIWER:

  1. The author should also report Raman spectroscopy of their synthesized materials, because it can show the number of layers and also size and defects in their structure.

AUTHORS:

As requested, the Raman spectrum of GO is also provided in the revision (Figure S2 of supporting information). It should be noted that the shape and position of the 2G band in defective graphenes is not as defined as in CVD defectless graphene and, therefore, the thickness of GO platelets measured by AFM is a safer technique to determine the number of layers. A comment on this is now made in the revised version (line 203-207).

REVIEWER:

  1. Authors provided AFM profile of their material but they didn’t provide which flake they did the measurement on.

AUTHORS:

Figure 1 shows a representative image of the GO sample used in this study. Over the years many groups, including ours have shown the easy exfoliation of GO. This point has now been clarified in the revision.

REVIEWER:

  1. The authors mentioned they did the XPS spectroscopy on single-layer graphene and represent the high resolution C1s for this. XPS analysis can be done on powder or films at least with couple of micrometer thickness. I wonder how they manage to perform XPS on single layers.

AUTHORS:

The reviewer is correct that XP spectroscopy can determine the elemental analysis of powders and films. In the present case, drops of aqueous solution of exfoliated GO are deposited on high quality fused quartz. After evaporation, the quartz plate is introduced in the vacuum chambers. According to AFM, this procedure leads mainly to the deposition of single layer GO. A comment on this has now been made in the experimental section of the revised version (line 194-197).

REVIEWER:

  1. To find C/O ratio from C1s high resolution XPS is an ambiguous practice. Authors should include survey spectrum of their samples and measure C/O ratio from it.

AUTHORS:

As requested by the reviewer the survey XPS is now provided in the supporting information (Figure S1). The C/O atomic ratio measured by the survey or high resolution peak areas were coincident, since the signal of the two peaks is large. A comment on this has now been introduced in the revision (line 198-199).

REVIEWER:

  1. Authors should edit image 2-d because it is assigned as “a”.

AUTHORS: Picture 2-d has been revised

REVIEWER:

  1. The authors should present tabulated results of their ICP-OES analysis.

AUTHORS:

C, N and S proportions were determined by elemental combustion analysis. The O content was determined by XPS. The Mn content of the two samples was determined by ICP-OES. These data are provided in page 1 of the supporting information.

“Combustion chemical analysis indicates that the carbon and sulphur content of the material was 54 and 4.2 %, respectively. Manganese and iron content determined by ICP-optical emission spectroscopy (ICP-OES) was 160 and 10 ppm, respectively.”

REVIEWER:

  1. Figure 2-c scale is not visible.

AUTHORS:

The scale bar has been specified in the caption and it was made more visible in the picture

REVIEWER:

  1. Authors claimed to measure WCA for their samples but they didn’t include any images of their samples.

AUTHORS:

We added in Supplementary information file four representative contact angle images of M1 and M7 membranes

Reviewer 2 Report

This manuscript reports the fabrication of PVDF-GO membranes by employing TEP as a solvent. Membranes were characterized by SEM, AFM, pore size, porosity, contact angle and mechanical tests and finally tested for photocatalytic methylene blue (MB+) degradation under visible light irradiation. The effect of different pH values of dye aqueous solutions on the photocatalytic efficiency was investigated. Salt effect assessment in removal efficiency was also tested. The obtained results sound good for preparing membranes in photocatalytic and there has potential for other applications such as distillation, solar to steam generation system, so on. However, it should be further improved before considering publication in the Journal.

  1. English should be revised through the manuscript
  2. The defined number affiliation should be used superscript
  3. Various words have been not used. Please remove it in the text.
  4. The abbreviations should be defined before use
  5. Other applications such as distillation, solar to steam generation system ( Power Sources, 2020, 448, 227388, Nano Energy, 2020, 68, 104324; Global Challenges 2018, 2, 1700094; Science & Technology Development Journal 2020, 23, 490), Electromagnetic shielding effectiveness (Applied Surface Science 2018, 435, 7) should be mentioned in the introduction.
  6. The quality of Figures is very low. Please improve it
  7. Page 2, line 63 authors mentioned on different procedures have been applied for introducing the photocatalyst. Other methods dry plasma reduction (J. Mater. Chem. A 2013, 1, 4436), liquid plasma reduction (Journal of Materials Chemistry 2012, 22, 14023; Solar Energy 2019, 191, 420; Synthetic Metals, 2020, 260, 116299; Materials Today Energy 2020, 16, 100384), thermal decomposition method (Journal of Electroanalytical Chemistry 2020, 857, 113769) should be referred.
  8. AFM measurement, please detail how to prepare the sample?
  9. TEM image, various nanoparticles were obtained. How to explain? If incorrect, please correct the image
  10. To confirm the presence of sulfur or other elements on the GO, please provide XPS survey, TEM-EDS, or SEM-EDS
  11. Figure 6, what are the nanoparticles? How to form?
  12. The formation of a porous, spongy-like structure should be explained. Please see Polymer, 2016, 101, 184; Chemistry-A European Journal 2018, 24, 561; Journal of Power Sources 2018, 376, 41 as references.

This manuscript can be considered for publication only when the above-mention questions were especially stressed in the revised manuscript. The referee would like to review a revised version of this paper in the future.

Author Response

This manuscript reports the fabrication of PVDF-GO membranes by employing TEP as a solvent. Membranes were characterized by SEM, AFM, pore size, porosity, contact angle and mechanical tests and finally tested for photocatalytic methylene blue (MB+) degradation under visible light irradiation. The effect of different pH values of dye aqueous solutions on the photocatalytic efficiency was investigated. Salt effect assessment in removal efficiency was also tested. The obtained results sound good for preparing membranes in photocatalytic and there has potential for other applications such as distillation, solar to steam generation system, so on. However, it should be further improved before considering publication in the Journal.

REVIEWER:

  1. English should be revised through the manuscript

AUTHORS:

Thanks for the comment. We revised all the manuscript correcting possible language mistakes.

REVIEWER:

  1. The defined number affiliation should be used superscript

AUTHORS:

Affiliation numbers have been indicated as superscripts

REVIEWER:

  1. Various words have been not used. Please remove it in the text.

AUTHORS:

We article has been revised and the mistakes have been corrected

REVIEWER:

  1. The abbreviations should be defined before use

AUTHORS:

All the abbreviations have been now defined in the manuscript.

REVIEWER:

5.Other applications such as distillation, solar to steam generation system ( Power Sources, 2020, 448, 227388, Nano Energy, 2020, 68, 104324; Global Challenges 2018, 2, 1700094; Science & Technology Development Journal 2020, 23, 490), Electromagnetic shielding effectiveness (Applied Surface Science 2018, 435, 7) should be mentioned in the introduction.

AUTHORS:

Thanks for the comment. The suggested and pertinent articles have been added to the manuscript in the introduction (line 30-32).

REVIEWER:

  1. The quality of Figures is very low. Please improve it

AUTHORS:

We tried to further improve the quality of the figures in order to make them more readable.

REVIEWER:

  1. Page 2, line 63 authors mentioned on different procedures have been applied for introducing the photocatalyst. Other methods dry plasma reduction (J. Mater. Chem. A 2013, 1, 4436), liquid plasma reduction (Journal of Materials Chemistry 2012, 22, 14023; Solar Energy 2019, 191, 420; Synthetic Metals, 2020, 260, 116299; Materials Today Energy 2020, 16, 100384), thermal decomposition method (Journal of Electroanalytical Chemistry 2020, 857, 113769) should be referred.

AUTHORS:

We thank the reviewer for the comment. However, the articles suggested refer to some techniques for the incorporation of catalysts in solar panels and photovoltaic devices and not in membranes that is the focus of our manuscript.

The manuscript section the reviewer refers to in fact states: “Different procedures have been applied for introducing the photocatalyst either on the membrane surface or inside the matrix.”

 In this regard, since we found some articles dealing with the use of plasma for the catalysts deposition on membrane surface, this technique has been added to the manuscript citing the following reference:

“These methods mainly include dip coating, photochemical method [32] layer-by-layer assembly [33,34], physical deposition [35], phase inversion [35], biaxial stretching [36], plasma reduction [37].

REVIEWER:

  1. AFM measurement, please detail how to prepare the sample?

The following details have been added into the relative supporting information section:

AUTHORS:

“GO membranes surfaces were mounted on a holder stub using double-sided scotch tape and they were imaged in a scan size of 2 µm x 2 µm.”

REVIEWER:

  1. TEM image, various nanoparticles were obtained. How to explain? If incorrect, please correct the image

AUTHORS:

TEM image of GO is shown in Figure 2 c. The typical 2D morphology with low contrast and exhibiting wrinkles reported for GO was recorded. The scale bar of the image corresponds to 500 nm and shows that the lateral size of the sheet is about 3 microns. At this magnification no nanoparticles are apparent in the image. However, it should be commented that all the samples were manipulated under conventional ambient and not in a clean room. It is known that dust and contaminants can be present and frequently observed under these conditions. A comment has now been added in the revision to note that the work out of the samples was performed under conventional ambient and not in a clean room and that dust particles could be present.

REVIEWER:

  1. To confirm the presence of sulfur or other elements on the GO, please provide XPS survey, TEM-EDS, or SEM-EDS

As requested by the reviewer’s the XPS survey spectrum of GO is now provided in the supporting information (Fugure S1)

REVIEWER:

  1. Figure 6, what are the nanoparticles? How to form?

We thank the referee for the comment. The GO nanoparticles of the membrane M10 shown in Figure 6 are clearly visible in the following Figure 7b (where the SEM of the same membrane M10 is shown). The reason why the nanoparticles are not visible in Figure 6 lies in the fact that the SEM pictures where taken at low magnification (10.000 x) for a better comparison among all the membranes. In Figure 7 the SEM magnification was increased (30.000 x) in order to make the GO nanoparticles visible. We better clarified this aspect into the manuscript:

“The presence of GO nanosheets in the M8 and M10 can be clearly observed in the images provided in Figure 7 taken at higher magnification.”

REVIEWER:

  1. The formation of a porous, spongy-like structure should be explained. Please see Polymer, 2016, 101, 184; Chemistry-A European Journal 2018, 24, 561; Journal of Power Sources 2018, 376, 41 as references.

We thank the reviewer for the comment. The formation of the sponge-like structure has been explained in the manuscript (lines 262-268; lines 234-237):

“The exposure time to humidity (from 2.5 to 5 min) greatly affected the morphology of the membranes prepared with additives (Figure 4 and 5). The exposure of the cast film to humid environment promoted the formation of a more porous structure on the surface while, along the cross-section, favored the formation a spongy architecture. The porogen effect of humidity is a quite well studied phenomenon and it is related to the adsorption, by the cast film, of water molecules from the humid air which delays the phase-inversion process fostering the creation of a porous and sponge-like structure [25,62–65].”

“It is well known, for example, that the presence of water in vapour form during VIPS step, promotes a delayed demixing, thus, favoring the formation of more porous, sponge-like structure in comparison with NIPS process. However, it should be mentioned that, during polymer precipitation, also kinetic factors play a relevant role for the polymeric chains arrangement.”

We included the reference suggested by the referee “Polymer, 2016, 101, 184”. The other proposed references are in authors’ opinion, not fully pertinent since they do not focus on the porogen effect of humidity on membranes. However,  the manuscript refers to other references (reference 25,62,63,64) which can support the results presented in this work.

Round 2

Reviewer 1 Report

In Figure S1, please specify elements accordingly, which peaks belong to carbon, oxygen etc.

In Figure S2, please specify D,G, 2D peaks. why there are no 2D peaks in your Raman spectra! ( should be around 2600 but it appears around 2900) 

It is an arbitrary unit (a.u.) not (u.a.)

In Figure S3, please show contact angels.

Author Response

Comments and Suggestions for Authors

1. REVIEWER:

In Figure S1, please specify elements accordingly, which peaks belong to carbon, oxygen etc.

AUTHOR:

Figure S1 has been edited to indicate the position of the C1s (283-290 eV) and O1s (530-536 eV) peaks. The reviewer is correct that in the survey spectrum is difficult to see the peaks corresponding to C and O. These peaks are recorded in high resolution XPS as presented in Figure 2d. A comment on this has been made.

2. REVIEWER:

In Figure S2, please specify D,G, 2D peaks. why there are no 2D peaks in your Raman spectra! ( should be around 2600 but it appears around 2900) 

AUTHOR:

The reviewer is correct. The 2900 cm-1 peak corresponds to the G+D peak. The assignments have been made in Figure S2 and a comment on the lack of resolution for the 2D peak is made in the main text.

3. REVIEWER:

It is an arbitrary unit (a.u.) not (u.a.)

AUTHOR:

The error has been corrected.

4. REVIEWER:

In Figure S3, please show contact angels.

AUTHOR:

The contact angle values have been added to the figure

Reviewer 2 Report

In general, the manuscript has been improved. However, it should be further improved before considering publication in the Journal.

  1. English should be revised through the manuscript.
  2. Various words have been not used. Please remove it in the text.
  3. As the AFM measurement, the thickness of GO is around 1 nm. However, the text is 1.5 nm. Please can you check it out?
  4. In TEM image, the explanation is not satisfying. If YES, authors should be carefully reconsidered other results. If NO, it should be better to remove TEM image and discussion.
  5. The authors referred to Figure S1, S2 in Supporting Information. However, the referee could not find any Figures. Is there any wrong during the upload file?
  6. There is not page and volume in the updated references

This manuscript can be considered for publication only when the above-mention questions were especially stressed in the revised manuscript. The referee would like to review a revised version of this paper in the future.

Author Response

Comments and Suggestions for Authors

In general, the manuscript has been improved. However, it should be further improved before considering publication in the Journal.

REVIEWER:

  1. English should be revised through the manuscript.

AUTHOR:

Minor English mistakes have been revised

REVIEWER:

2. Various words have been not used. Please remove it in the text.

AUTHOR:

Some words have been deleted as requested.

REVIEWER:

3. As the AFM measurement, the thickness of GO is around 1 nm. However, the text is 1.5 nm. Please can you check it out?

AUTHOR:

The thickness of single layer GO is around 1 nm. The mismatch has been corrected in the text.

REVIEWER:

4. In TEM image, the explanation is not satisfying. If YES, authors should be carefully reconsidered other results. If NO, it should be better to remove TEM image and discussion.

AUTHOR:

TEM image clearly shows sheets of GO all over the field, with a very light dark contrast. The TEM image has been maintained and the text revised.

REVIEWER:

5. The authors referred to Figure S1, S2 in Supporting Information. However, the referee could not find any Figures. Is there any wrong during the upload file?

AUTHOR:

By mistake has been uploaded an old version of the supporting information file. Now the file has been updated all the figures are visible

REVIEWER:

6. There is not page and volume in the updated references

AUTHOR:

Thanks for the comment. All the references have been updated.
